# Capture at the ER-mitochondrial contacts licenses IP$_3$ receptors to stimulate local Ca$^{2+}$ transfer and oxidative metabolism

Máté Katona [1], Ádám Bartók [1], Zuzana Nichtova[1], György Csordás[1], Elena Berezhnaya[1], David Weaver[1], Arijita Ghosh[1], Péter Várnai [2], David I. Yule [3] & György Hajnóczky [1] ✉

Endoplasmic reticulum-mitochondria contacts (ERMCs) are restructured in response to changes in cell state. While this restructuring has been implicated as a cause or consequence of pathology in numerous systems, the underlying molecular dynamics are poorly understood. Here, we show means to visualize the capture of motile IP$_3$ receptors (IP3Rs) at ERMCs and document the immediate consequences for calcium signaling and metabolism. IP3Rs are of particular interest because their presence provides a scaffold for ERMCs that mediate local calcium signaling, and their function outside of ERMCs depends on their motility. Unexpectedly, in a cell model with little ERMC Ca$^{2+}$ coupling, IP3Rs captured at mitochondria promptly mediate Ca$^{2+}$ transfer, stimulating mitochondrial oxidative metabolism. The Ca$^{2+}$ transfer does not require linkage with a pore-forming protein in the outer mitochondrial membrane. Thus, motile IP3Rs can traffic in and out of ERMCs, and, when 'parked', mediate calcium signal propagation to the mitochondria, creating a dynamic arrangement that supports local communication.

Local communication between intracellular organelles at membrane contacts is a burgeoning topic in cell biology and medicine. The contacts between the endoplasmic reticulum (ER) and mitochondria (ERMCs) have been revealed as central to a broad range of functions, including lipid biosynthesis, calcium signaling, cell survival regulation, and organelle dynamics and turnover. ERMCs have been documented in many different cell types and tissues, and compelling evidence supports their broad health relevance[1–5]. Moreover, the studies of ERMCs have served as a template for studies of other interorganellar junctions[6,7].

ERMCs are supported by protein tethers which show varying lengths and shapes[8]. Dozens of proteins have been implicated in ERMC tethering[7,9]. The creation of synthetic linkers has allowed perturbation and measurement of contacts and local communication in live cells, and these studies have provided direct evidence for local Ca$^{2+}$ and reactive oxygen species signaling at ERMCs[10–12]. It has also been shown that ERMCs tighten in response to various forms of cellular stress[8] and that mobile mitochondria are retained close to the ER during cytoplasmic [Ca$^{2+}$] ([Ca$^{2+}$]$_c$) elevations and might locally support the energy supply and [Ca$^{2+}$]$_c$ buffering[13]. Thus contacts seem to dynamically restructure to meet the changing demands of the cell, but how they are formed and dissolved is yet to be determined.

One of the most investigated inter-organelle communication mechanisms is the inositol 1,4,5-trisphosphate receptor (IP3R) mediated Ca$^{2+}$ transport from the ER to the mitochondria that results in the mitochondrial matrix [Ca$^{2+}$] ([Ca$^{2+}$]$_m$) signals to stimulate ATP production and determine cell fate. IP3Rs are IP$_3$-and Ca$^{2+}$-regulated Ca$^{2+}$ release channels of the ER that mediate intracellular Ca$^{2+}$ mobilization and that, together with the ensuing store-operated Ca$^{2+}$ entry (SOCE), give rise to [Ca$^{2+}$]$_c$ signals upon stimulation of cells by a wide range of

[1]MitoCare Center, Department of Pathology, Anatomy and Cell Biology, Thomas Jefferson University, Philadelphia, PA, USA. [2]Department of Physiology, Semmelweis University, Budapest, Hungary. [3]Department of Physiology and Pharmacology, University of Rochester, Rochester, NY, USA. ✉e-mail: gyorgy.hajnoczky@jefferson.edu

hormones and neurotransmitters. IP3R channel function is also modulated by ATP, phosphorylation, redox, and by a large number of interacting proteins, including IRBIT, Bcl-2, and Bcl-x$_L$. There are three distinct isoforms of IP3R, at least one of which is expressed in almost all cell types. The different IP3R isoforms show heterogeneity in tissue and intracellular distribution, structure, and regulation by IP$_3$, Ca$^{2+}$, and other factors, but have similar Ca$^{2+}$ conductances and gating kinetics[14–18].

IP$_3$-induced [Ca$^{2+}$]$_c$ transients are propagated to the mitochondria through the mitochondrial Ca$^{2+}$ uniporter (mtCU). During IP3R-mediated Ca$^{2+}$ release from the ER, a [Ca$^{2+}$] nanodomain, which can be an order of magnitude higher than the bulk [Ca$^{2+}$]$_c$, is generated at the ERMCs[10,19]. This Ca$^{2+}$ signal crosses the outer mitochondrial membrane (OMM), presumably through the abundant β-barrel forming OMM proteins (VDACs, TOM40, SAM50), into the intermembrane space (IMS). Ca$^{2+}$ entry into the matrix through mtCU, driven by the inside negative IMM potential (ΔΨ$_m$), is steeply dependent on [Ca$^{2+}$]$_{IMS}$ to effectively filter out small fluctuations while responding robustly to larger [Ca$^{2+}$] elevations[20,21].

Previous studies suggest that IP3Rs are not only needed for Ca$^{2+}$ fluxes but also participate in forming close ERMCs via interaction with OMM proteins. IP3R1 and IP3R3 have been shown to co-immunoprecipitate with the OMM resident VDAC1 (refs. 22, 23, 24) and evidence has been presented for a multi-protein complex among IP3Rs, VDAC1 and GRP75 (ref. 22) and TOM70 (Filadi et al. 2018) facilitating the propagation of Ca$^{2+}$ transients to the mitochondrial matrix. In terms of the ERMC structure, loss of IP3Rs by knockout has been shown to reduce the close contact both in chicken B cells (DT40) and the human cervical cancer cell line (HeLa). In DT40 cells, the loss of close contacts was completely rescued by the reintroduction of any single IP3R isoform, independent of its function as a Ca$^{2+}$ channel[25]. Notably, IP3R2 showed an advantage over IP3R1 and IP3R3 in populating ERMC and mediating the ER-mitochondrial Ca$^{2+}$ transfer[25]. However, the respective role of each IP3R isoform in the context of ERMC Ca$^{2+}$ transfer has never been systematically tested in mammalian cells.

IP3Rs exhibit both immobile and motile states. Clusters of immobile IP3R1s have been shown to respond to IP$_3$/Ca$^{2+}$ signaling stimuli close to the plasma membrane (PM), whereas motile IP3R were found not to release Ca$^{2+}$[26]. To transition to the immobile and Ca$^{2+}$ release competent state, the IP3R1 undergoes a 'licensing' procedure[27]. Recent evidence indicates that KRAS-induced actin-interacting protein (KRAP) determines which IP3R1s respond by tethering IP$_3$R clusters to actin alongside sites where SOCE occurs, licensing them to evoke Ca$^{2+}$ puffs and global [Ca$^{2+}$]$_c$ signals. These data implicate the actin cytoskeleton in IP3R regulation[28]. Interestingly, actin polymerization has also been suggested to play a role in ER-mitochondrial Ca$^{2+}$ transfer[29], but dynamic visualization or perturbation of IP3Rs at ERMCs has been difficult to achieve.

We, here, tested each IP3R isoform's coupling to the mitochondria in mammalian, HEK293 (HEK) cells that inherently display limited ER-mitochondrial Ca$^{2+}$ transfer[30]. After establishing the ERMC phenotype associated with the rescue of each IP3R isoform in IP3R-deficient deficient HEK cells, we utilized a synthetic linker strategy to trap the mobile IP3Rs close to the mitochondria and study IP3R dynamics in the context of [Ca$^{2+}$]$_m$ and oxidative metabolism.

## Results

### IP3R-dependence of mitochondrial Ca$^{2+}$ signaling and bioenergetics in HEK cells

We studied mitochondrial Ca$^{2+}$ handling in wild-type HEK cells (WT HEK) and in a clone lacking all three IP3R isoforms (TKO)[31]. Cells were transfected with a mitochondrial matrix-targeted Ca$^{2+}$ sensor Cepia3 (mtCepia3) and loaded with Fura-2/AM to monitor [Ca$^{2+}$]$_m$ simultaneously with [Ca$^{2+}$]$_c$. Upon addition of an IP$_3$-generating agonist,

carbachol, WT cells responded with a rapid [Ca$^{2+}$]$_c$ increase which was followed by a modest increase in [Ca$^{2+}$]$_m$ (Fig. 1A). Using higher affinity matrix-targeted Ca$^{2+}$ sensor GCaMP6f the [Ca$^{2+}$]$_m$ increase was more visible but partial mistargeting of these probes to the cytosol was also more apparent (Fig. S1B). This weak [Ca$^{2+}$]$_c$-[Ca$^{2+}$]$_m$ coupling during IP3R-linked stimulation distinguishes HEK cells from HeLa cells (Fig. S1A) and several other cell lines[30]. In the TKO cells, no [Ca$^{2+}$]$_c$ or [Ca$^{2+}$]$_m$ response was evoked by agonist stimulation, confirming the effective knockout of IP3Rs (Fig. 1A).

We next studied whether the loss of IP3Rs affected the mitochondria's ability to take up calcium. By western blot, we observed no changes in the protein abundance of the components of the mtCU, the pore-forming protein, MCU, the scaffold, EMRE, or the Ca$^{2+}$ sensing regulators, MICU1 and MICU2 (Fig. 1B). Measuring mitochondrial Ca$^{2+}$ uptake in permeabilized cells with a fluorometric Ca$^{2+}$ clearance assay, we also found no difference between WT and TKO, with or without the mtCU inhibitor Ruthenium Red (RuRed) (Figs. 1C and S1D), nor was the ΔΨ$_m$ altered, as measured by tetramethylrhodamine ethyl ester (TMRE) quenching (Fig. S1C).

To assess the structural context of the local ER to mitochondria Ca$^{2+}$ transfer, we analyzed the fraction of the mitochondrial surface involved in ERMCs (0–100 nm) by transmission electron microscopy (TEM). We observed close ER-mitochondrial associations in both WT and TKO cells, however, TKO cells had fewer close contacts in the 0–20 nm range (Fig. 1D), consistent with results from IP3R TKO DT40 and HeLa cells[25]. Thus, WT HEK cells display weak ER-mitochondrial Ca$^{2+}$ coupling, despite having (1) IP3R-mediated ER Ca$^{2+}$ release, (2) IP3R-dependent ER-mitochondrial contacts with the gap width relevant for Ca$^{2+}$ transfer, and (3) effective mitochondrial Ca$^{2+}$ sequestration. The IP3R TKO cells provide a suitable platform for testing if the ER-mitochondrial Ca$^{2+}$ coupling can be manipulated by the expression of any individual IP3R isoform.

### ER-mitochondrial Ca$^{2+}$ transfer is weakly supported by overexpressed IP3Rs in TKO, showing a preference for IP3R2

HEK cells express all three IP3R isoforms, which are differently sensitive to their main regulators IP$_3$ and Ca$^{2+}$[31] and different in their ability to tether and communicate with mitochondria[25]. By re-expressing IP3Rs in the TKO cells, we studied the involvement of each individual isoform on ER-mitochondrial Ca$^{2+}$ signal propagation. Acute rescue of TKO cells with each IP3R isoform completely rescued the agonist-induced [Ca$^{2+}$]$_c$ responses at 48 h, but the Ca$^{2+}$ signal propagation to the mitochondria, measured by mtCepia3 remained weak, similar to WT (Fig. 2A upper). We tested if longer overexpression would increase the abundance of the IP3Rs and/or allow the formation of ERMCs supporting Ca$^{2+}$ propagation (Fig. 2A lower). After 120 h of overexpression, the peak [Ca$^{2+}$]$_c$ response to carbachol increased in IP3R1 rescue (48 h $0.58 \pm 0.03$ μM vs 120 h $0.74$ μM $\pm 0.05$, $p < 0.001$), while there was no significant change in [Ca$^{2+}$]$_c$ transients in the IP3R2 and IP3R3 rescue (Fig. 2A). However, all rescue conditions had enhanced [Ca$^{2+}$]$_m$ responses at 120 h (Fig. 2A bottom).

We studied the rapid propagation of the [Ca$^{2+}$]$_c$ signal into the mitochondria by plotting the [Ca$^{2+}$]$_m$ at the time of the [Ca$^{2+}$]$_c$ peak as a function of the corresponding peak [Ca$^{2+}$]$_c$ values. A subset of cells had a fast [Ca$^{2+}$]$_m$ increase, defined as [Ca$^{2+}$]$_m$ half peak within 3 s of the [Ca$^{2+}$]$_c$ half peak (Fig. 2B, triangles in the highlighted area). With 48 h overexpression, only expression of IP3R2 resulted in a few cells with fast Ca$^{2+}$ transfer (3 out of 24; the time course for these cells is in Fig. S2A), whereas at 120 h, some cells expressing each of the three isoforms had close coupling between [Ca$^{2+}$]$_c$ and [Ca$^{2+}$]$_m$ (Fig. 2C, top). In the case of the IP3R2 rescue, we observed that most cells with a fast [Ca$^{2+}$]$_m$ response did not have a high [Ca$^{2+}$]$_c$ peak (Fig. 2B, see the red area (IP3R2) to the left of the green (IP3R1) and blue (IP3R3) areas). This representation of the data highlighted that the early [Ca$^{2+}$]$_m$ events

mediated by IP3R2 were not dependent on the magnitude of the global $[Ca^{2+}]_c$ rise but were the result of local $Ca^{2+}$ signal propagation from IP3R2 to the mitochondria. We also determined the expression levels of IP3Rs and mitochondrial uniporter complex components in WT, TKO, and IP3R overexpressing conditions. Immunoblots confirmed the endogenous expression of the three IP3R isoforms in WT and their absence in the TKO. All rescue conditions showed a time-dependent increase in IP3R protein levels during acute re-expression, which was several-fold higher than the endogenous protein levels in the case of IP3R1 and IP3R3. IP3R2 rescue had lower protein levels after 48 h, and at 120 h, showed slight overexpression compared to the WT (Fig. 2D). No changes were observed in the levels of MCU or its auxiliary proteins across the different IP3R rescue conditions (Fig. S2C, D), suggesting that the increase in $Ca^{2+}$ coupling is not caused by changes in mitochondrial $Ca^{2+}$ uptake.

It is worth noting that, with time, more cells had $[Ca^{2+}]_m$ responses in each rescue condition, however, they likely reached peak $[Ca^{2+}]_m$ due to the sustained global $[Ca^{2+}]_c$ signal, rather than local ER-mitochondrial $Ca^{2+}$ communication. We calculated the first derivatives of the $[Ca^{2+}]_c$ and $[Ca^{2+}]_m$ time courses from the selected cells to compare the cytoplasmic and mitochondrial $Ca^{2+}$ rates of rise (Fig. 2C bottom) and found that the dynamics of the agonist-induced $Ca^{2+}$ responses in the cytosol were similar for each isoform (Fig. S2E, left). IP3R1 gave the highest rates of $[Ca^{2+}]_m$ change, though the initiation of the mitochondrial $Ca^{2+}$ rise more closely followed the cytosolic rise with IP3R2 (Fig. 2C bottom, "Coupling time"). The greater rate of $[Ca^{2+}]_m$ rise in the case of IP3R1, likely reflects the prolonged cytosolic increase, which achieves a relatively high maximal $[Ca^{2+}]_c$, rather than ER-mitochondrial coupling (Fig. S2E, right). We also used clonal stable rescues of HEK-TKO cells with each IP3R

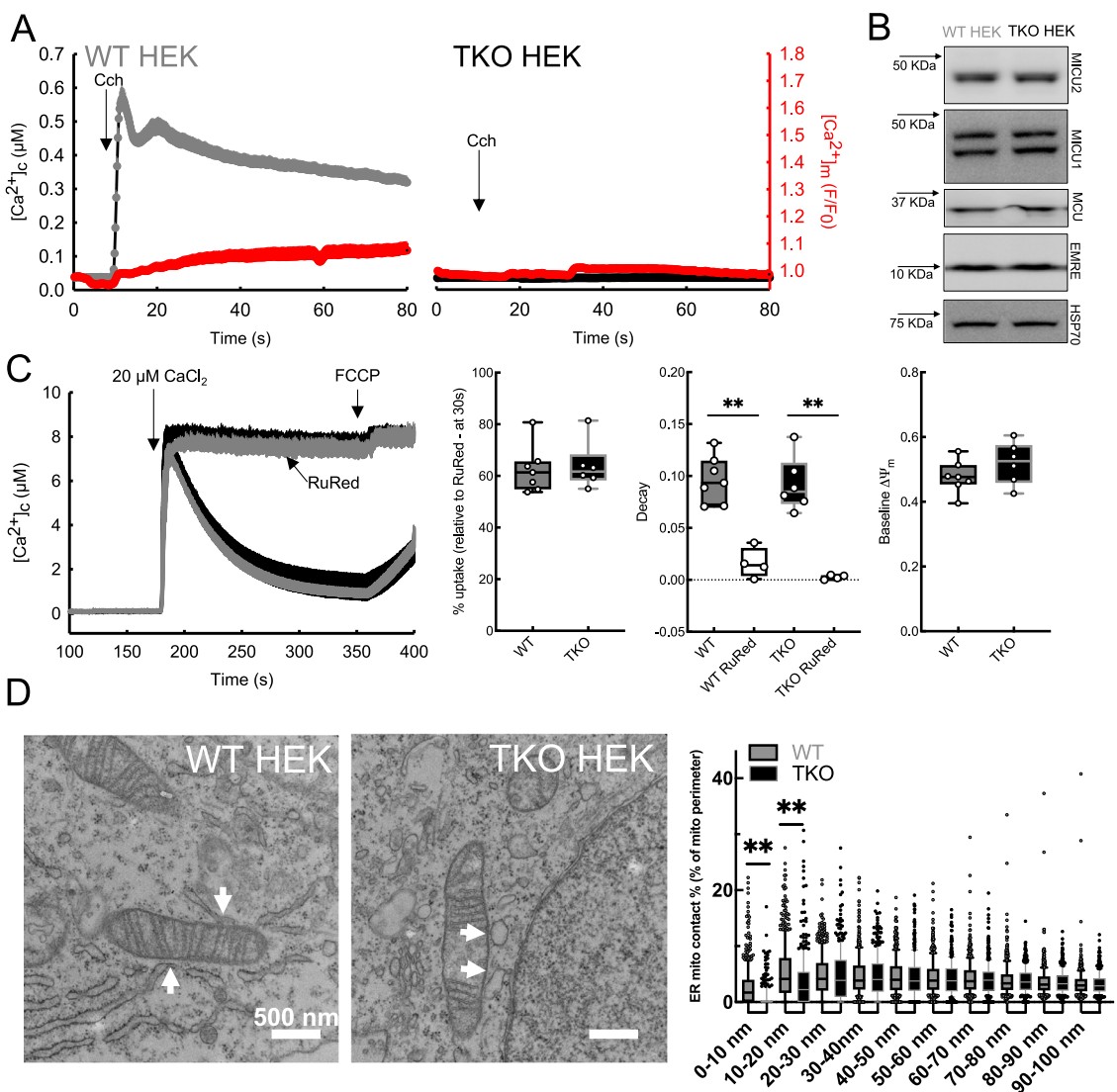

**Fig. 1 | HEK cells have weak ER-mitochondrial Ca²⁺ transfer, but tight and IP3R-dependent ERMCs. A** Average $[Ca^{2+}]_c$ response of WT (left, gray; $n = 25$) and TKO (right, black; $n = 15$) HEK cells to Carbachol (Cch) and the corresponding changes in the $[Ca^{2+}]_m$ (red traces). **B** Immunoblots showing the protein levels of MCU, MICU1, MICU2, and EMRE in WT and TKO HEK lysates. **C** Mean mitochondrial $Ca^{2+}$ clearance by permeabilized WT and TKO cells after the addition of 20 μM CaCl₂ bolus in the presence of thapsigargin (2 μM) and CGP (20 μM) with or without RuRed treatment. FCCP was added at the end of every run to release the accumulated $Ca^{2+}$ from the mitochondria. Box plots show the RuRed sensitive $Ca^{2+}$ uptake, the rate of $Ca^{2+}$ clearance after the $Ca^{2+}$ pulse, and the baseline mitochondrial membrane potential.

(WT $n = 7$, TKO $n = 6$, WT + RuRed, TKO + RuRed $n = 4$, **$p \leq 0.001$, One-way ANOVA, Holm–Sidak method, box plots indicate median, 25th and 75th percentile (box), and 5th and 95th percentile (whiskers) and all datapoints). **D** Representative TEM images of WT (left) and TKO HEK (right) cells, with white arrows marking ERMCs. Box plots showing the percentage of mitochondrial perimeter within 0–100 nm distance to the ER using 10 nm bins in WT (gray) and TKO HEK (black). (WT $n = 455$, TKO $n = 377$ mitochondria from three independent fixation for each; **<0.001 Unpaired $t$-test, Two-tailed $p$ values, box plots indicate median, 25th and 75th percentile (box), and 10th and 90th percentile (whiskers) as well as outliers (dots)). Source data are provided for each panel as a Source Data file.

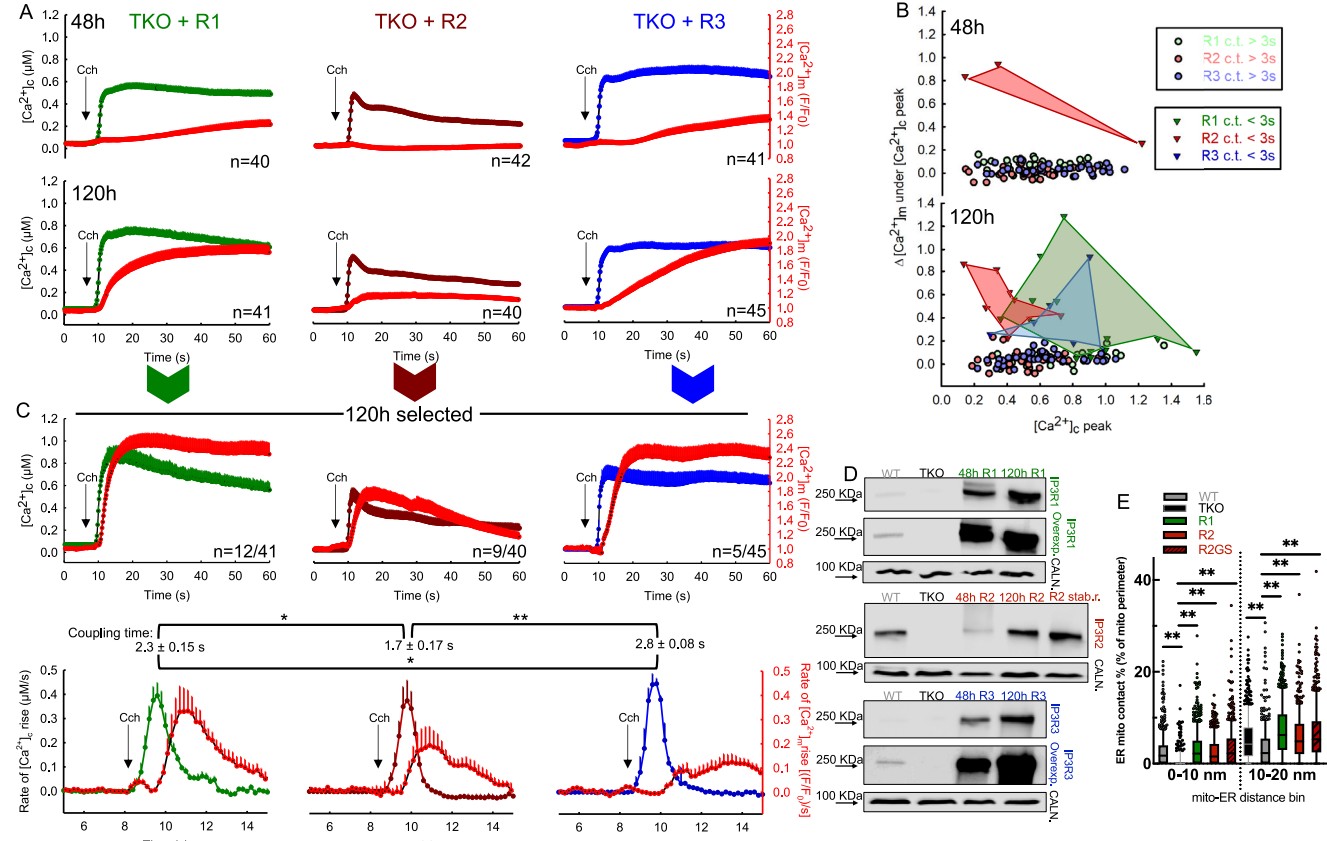

**Fig. 2 | IP3R overexpression in TKO cells enhanced ER-mitochondrial Ca²⁺ signaling and restored close ERMCs. A** Mean time courses of the Cch-induced $[Ca^{2+}]_c$ responses in R1 (green), R2 (maroon), and R3 (blue) acute 48 h (upper panel) and 120 h (lower panel) rescue in TKO HEK cells. Corresponding changes in the $[Ca^{2+}]_m$ are shown as red traces on each graph. **B** $[Ca^{2+}]_m$ at the time of $[Ca^{2+}]_c$ peak plotted as a function of the $[Ca^{2+}]_c$ peak in 48 h (upper panel) and 120 h (lower panel) rescue. The shaded area highlights the cells showing tight ERMC Ca²⁺ coupling. **C** Average $[Ca^{2+}]_c$ responses of a selected subpopulation of cells with less than 3 s cyto-mito coupling time after 120 h of expression of IP3R1 (green), R2 (maroon) and R3 (blue) and the corresponding changes in $[Ca^{2+}]_m$ (red) (upper panel). Mean traces of the cytosolic and corresponding mitochondrial rates of Ca²⁺ rise in the selected 120 h acute rescue TKO cells, and the calculated average coupling time for

each condition (±SEM, One-way ANOVA, normality test (Shapiro−Wilk), all pairwise multiple comparison procedures (Holm−Sidak method)). **D** Immunoblot showing the IP3R expression in WT HEK, TKO HEK, and IP3R acute rescue TKO cells for 48 and 120 h and TKO HEK cells stably rescued with R2. **E** Box plots showing the percentage of close, 0–10 and 10–20 nm contacts in WT (gray; $n = 455$ mitochondria/3 independent fixation), TKO (black; $n = 377/3$) and TKO HEK stably expressing R1 (green; $n = 464/3$), R2 (maroon; $n = 383/3$) and pore mutant R2GS3 (shaded maroon; $n = 411/3$) (**$p < 0.001$; Unpaired $t$-test; Two-tailed $p$ value, box plots indicate median, 25th and 75th percentile (box), and 10th and 90th percentile (whiskers) as well as outliers (dots)). Source data are provided for each panel as a Source Data file.

isoform to study the long-term consequences of IP3R overexpression on the ER-mitochondrial Ca²⁺ coupling and found that Ca²⁺ signal propagation to the mitochondria was similar to that in the 120 hr overexpression condition further supporting that the duration of IP3R expression was not a limiting factor in the local ERMC Ca²⁺ coupling (Fig. S2B).

Finally, we tested the ability of overexpressed IP3Rs to support local ERMCs in HEK cells by transmission electron microscopy. Stable reintroduction of IP3R1 or IP3R2 and even a non-Ca²⁺-conducting, "pore-dead" mutant of IP3R2 into TKO cells completely restored the close ER-mitochondrial contacts (Fig. 2E), consistent with our previous findings in DT40 cells[25].

In summary, re-expression of each individual IP3R isoform was sufficient to restore ER Ca²⁺ mobilization and close ERMCs, but none produced consistently robust ER-mitochondrial Ca²⁺ transfer, despite protein levels at least as high as in WT cells. TKO cells rescued with IP3R2 showed the highest frequency of tightly coupled cells even at protein levels comparable to WT than IP3R1 or IP3R3 rescues, and with relatively small $[Ca^{2+}]_c$ signals.

## Many IP3Rs are motile and can readily be trapped close to the mitochondria by drug-inducible synthetic linkers

We reasoned that the spatial distribution of IP3Rs limits their ability to transfer Ca²⁺ to the mitochondria in the HEK cells. To explore this possibility, we first assessed the mobility of the different IP3R isoforms using fluorescence recovery after photobleaching (FRAP). Specifically, we tested whether there are differences in the lateral mobility of different IP3R isoforms in the ER membrane and if the mobility is influenced by IP₃-generating agonist (carbachol) stimulation. TKO cells were rescued with mCherry-tagged IP3R1 or mRFP-tagged IP3R2 or IP3R3. After rapidly photobleaching a 5 μm diameter circular area, we monitored the recovery of fluorescence in the bleached area. Although all isoforms presented a remarkable recovery of fluorescence, IP3R1 and 3 had a slightly higher mobile fraction than IP3R2 (Fig. 3A, B). This difference was confirmed across expression levels (Fig. S3A). Carbachol stimulation affected neither the recovery rate nor the recovery kinetics of any isoform (Fig. 3A).

To specifically study IP3R dynamics close to the ERMCs we created a drug-inducible linker to capture IP3Rs that draw close to the OMM (see the design in Fig. 4A). We focused on IP3R3 because it

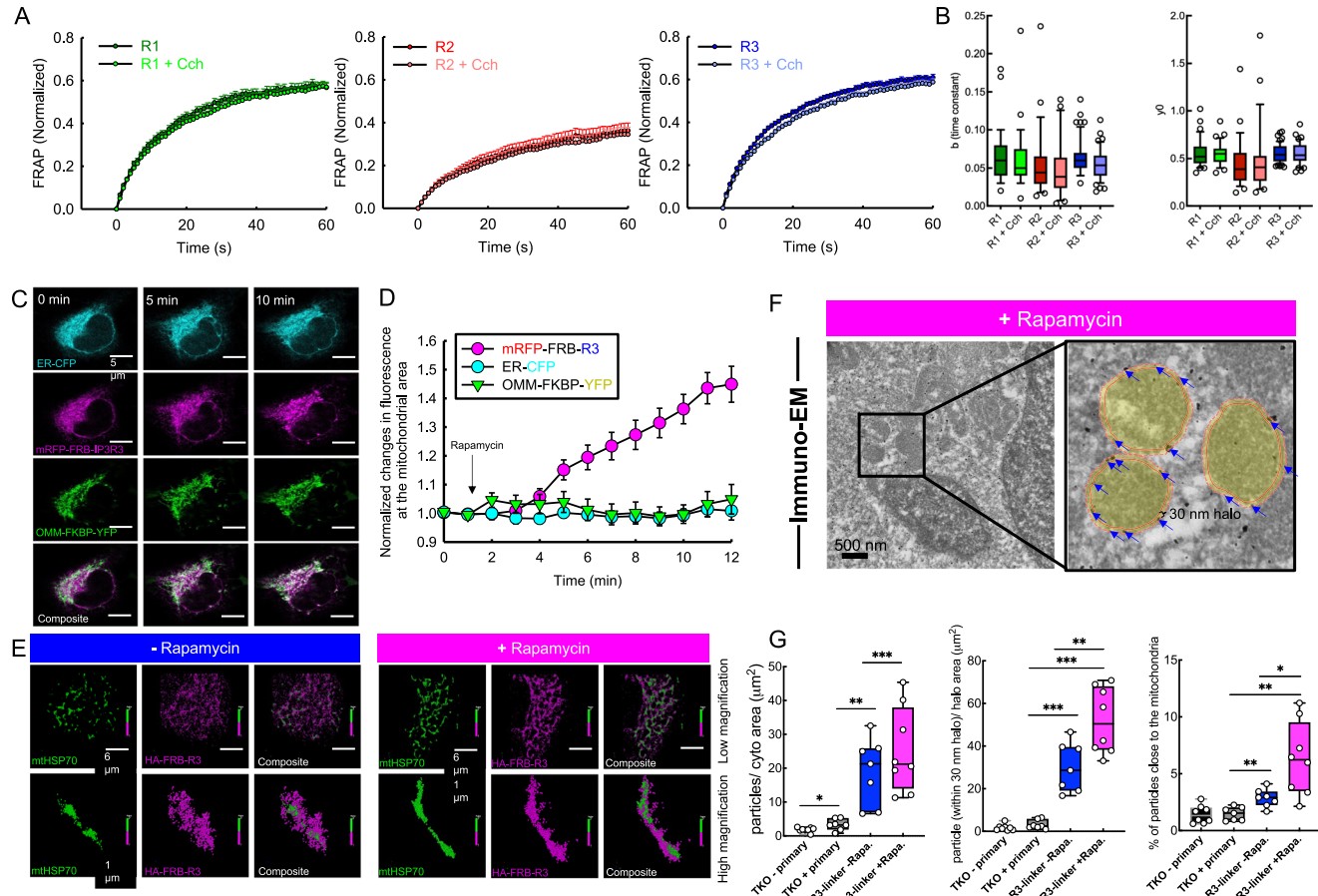

**Fig. 3 | Lateral mobility of IP3Rs allows their trapping at ERMCs. A** Average time courses showing IP3R-mRFP fluorescence recovery after photobleaching normalized to the pre-bleach intensity in unstimulated R1 (green), R2 (maroon), and R3 (blue) and Cch stimulated R1 (light green), R2 (light red) and R3 (light blue) acute TKO rescue cells (datapoints represent averages with ±SEM, (R1 $n = 32$, R1 + Cch $n = 37$, R2 $n = 37$, R2 + Cch $n = 23$, R3 $n = 55$, R3 + Cch $n = 43$ from three experiments). **B** Calculated time constant and recovery (y0) for each IP3R isoform with and without agonist stimulation (R1 $n = 32$, R1 + Cch $n = 37$, R2 $n = 25$, R2 + Cch $n = 23$, R3 $n = 55$, R3 + Cch $n = 43$ from three experiments; Unpaired $t$-test; Two-tailed $p$ value; b: R1 vs R1 + Cch $p = 0.4819$, R2 vs R2 + Cch $p = 0.4494$, R3 vs R3 + Cch $p = 0.0585$; y0: R1 vs R1 + Cch $p = 0.7404$, R2 vs R2 + Cch $p = 0.7596$, R3 vs R3 + Cch $p = 0.9114$, Box plots indicate median, 25th and 75th percentile (box), and 5th and 95th percentile (whiskers) and outliers (single points). **C** Representative images showing the R3-mRFP (magenta), ER-CFP (cyan), and OMM-YFP (YFP) signals before and after (5 and 10 min) the addition of rapamycin. **D** R3-mRFP-FRB (magenta circles) and ER-CFP (cyan circles) co-localization over time with FKBP-YFP-OMM (green triangles), presented by changes in mRFP and CFP fluorescence under the YFP mask over time. ($n = 10$ cells from three experiments; datapoints represent averages with ±SEM). **E** STORM super-resolution images showing the R3-OMM positioning without (before) and with (after) rapamycin treatment.

**F** Representative Immuno-EM image of a rapamycin-treated, R3-linker-expressing cell and a magnified field with highlighted mitochondria (yellow). The 30 nm halo area is between the solid and dashed red lines. Individual gold particles within the halo area are marked with blue arrows. **G** Quantification of the ImmunoGold labeling in non-transfected no primary (black; $n = 8$), non-transfected plus primary (dark gray; $n = 7$), IP3R3-linker without rapamycin (blue; $n = 7$) and R3-linker with rapamycin treatment (magenta; $n = 8$) (Images were processed from from three independent fixation, Statistical probe: Unpaired $t$-test; Two-tailed $p$ values, particles/cyto area: TKO - primary vs TKO + primary $p = 0.0396$, R3-linker -Rapa. vs +Rapa. $p = 0.369$, TKO + primary vs R3-linker -Rapa $p = 0.0012$, TKO + primary vs R3-linker +Rapa $p = 0.0005$; particles within 30 nm halo/halo area: TKO - primary vs TKO + primary $p = 0.1384$, R3-linker -Rapa. vs +Rapa. $p = 0.0091$, TKO + primary vs R3-linker -Rapa <0.0001, TKO + primary vs R3-linker +Rapa <0.0001; % of particles close to the mitochondria: TKO - primary vs TKO + primary $p = 0.9062$, R3-linker -Rapa. vs +Rapa. $p = 0.0166$, TKO + primary vs R3-linker -Rapa $p = 0.0031$, TKO + primary vs R3-linker +Rapa $p = 0.0019$* $p < 0.05$, ** $p < 0.01$, ***<0.0001 box plots indicate median, 25th and 75th percentile (box), and 10th and 90th percentile (whiskers) and all datapoints)). Source data are provided for panels **A**, **B**, **D**, **G** as a Source Data file.

showed the least effective Ca²⁺ transfer to the mitochondria and had a high level of lateral mobility. We evaluated the redistribution of IP3R3 in TKO cells expressing IP3R3-mRFP-FRB and OMM-FKBP-YFP during rapamycin-induced FRB-FKBP dimerization by high-resolution confocal microscopy. To monitor changes in the ER morphology, we expressed ER lumen-targeted CFP together with the linkers. We quantified the re-localization of IP3R3-mRFP by recording specific changes in the IP3R3-mRFP and ER-CFP fluorescence in the area colocalized with mitochondria using the YFP channel (OMM) as a mask (Fig. 3C and Supplementary Movie 1). Induction of FRB-FKBP linkage resulted in time-dependent IP3R3-mRFP co-localization with OMM-YFP while the ER-CFP fluorescence distribution remained unaltered,

suggesting that IP3R3 was captured at the ER-mitochondrial interface without overall altering the ER structure (Fig. 3D and Supplementary Movie 1).

We further studied IP3R3-OMM co-localization using super-resolution Stochastic Optical Reconstruction Microscopy (STORM) in fixed TKO cells expressing IP3R3-HA-FRB and OMM-FKBP with and without rapamycin-induced linkage (10 min). HA-tag was labeled with AF647 and the endogenous mtHSP70 (mitochondria) with CF568. Untreated cells showed a reticular-like distribution of IP3Rs, diffusely scattered around mitochondria, in contrast to the rapamycin-treated cells where IP3R3 tightly surrounded the OMM (Fig. 3E). We also used immuno-EM to study, at the high spatial resolution, the localization of

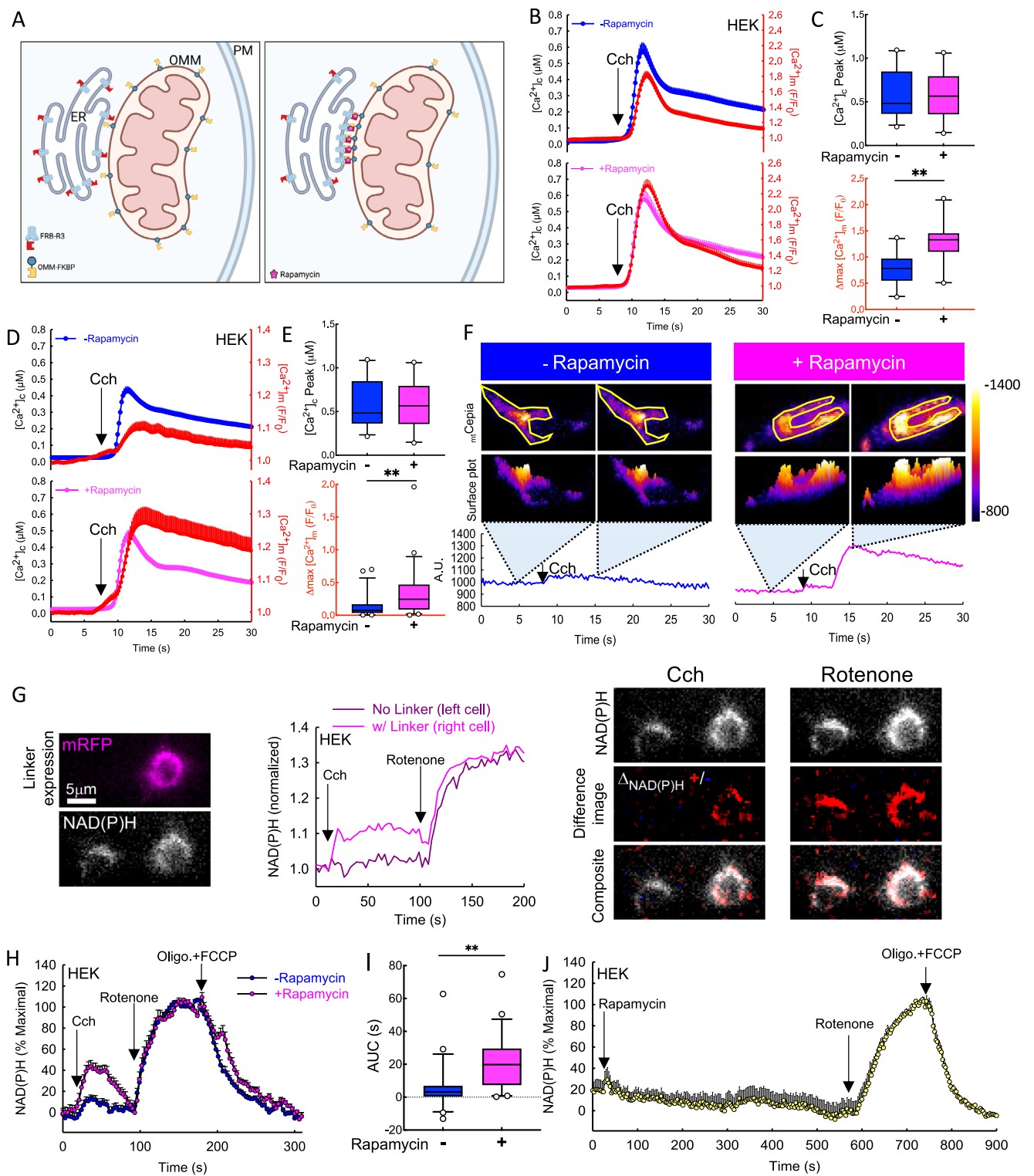

IP3R3 after rapamycin-induced linkage to the contacts. We detected the IP3Rs using an anti-HA antibody to label the FRB-IP3R3 constructs. IP3Rs tended to populate the close contacts at the 0–25 nm range[25]. Therefore, taking the size of the nanogold particles into consideration, we chose a range of 0–30 nm from the outline of the mitochondrion for IP3R detection (Fig. 3F). In agreement with the STORM imaging, we saw an increase in the anti-HA gold particles in the 0–30 nm of the mitochondria after 5 min of rapamycin treatment confirming rapid relocalization of mobile IP3Rs to the ERMCs (Fig. 3G). Thus, all IP3R isoforms had lateral mobility in the ER membrane and mobile IP3R3s

could be captured at the ERMCs by synthetic linkers without altering the ER structure.

## Capture of IP3R3 close to the mitochondria immediately enhances local Ca²⁺ transfer and Ca²⁺ sensitive matrix dehydrogenase activity

Previous reports indicated that immobile IP3R1 clusters at ER-PM junctions, and not motile IP3R clusters, release Ca²⁺ and that, IP3Rs require "licensing" to mediate Ca²⁺ fluxes upon immobilization[26,27]. Thus, the question arises: Can IP3Rs immobilized at the ERMCs

**Fig. 4 | Effective ER-mitochondrial Ca²⁺ communication is dependent on IP3R3 localization to the ERMCs. A** Scheme depicting R3-FRB and FKBP-OMM linkage by rapamycin. (Created in BioRender). **B** Average time courses showing the carbachol-induced changes in the $[Ca^{2+}]_c$ and the corresponding changes in $[Ca^{2+}]_{IMS}$ (red traces), without (blue) and with (pink) rapamycin treatment in R3-FRB and FKBP-OMM expressing TKO HEK cells in 0 mM Ca²⁺ ECM. **C** Box plots showing the maximal change in $[Ca^{2+}]_c$ (upper) and $[Ca^{2+}]_{IMS}$ (lower) for both conditions (−Rapa $n = 22$, +Rapa $n = 18$), Unpaired $t$-test; Two-tailed $p$ value; $[Ca^{2+}]_c$ peak $p = 0.9818$, $\Delta max[Ca^{2+}]_m$ **$p < 0.001$. **D** Mean $[Ca^{2+}]_c$ and corresponding $[Ca^{2+}]_m$ (red traces) responses to agonist stimulation under the same conditions as described in (**E**). Maximum $[Ca^{2+}]_c$ (upper) and $[Ca^{2+}]_m$ (lower) responses to agonist (**$p < 0.001$, Kruskal–Wallis One-way ANOVA on Ranks, all pairwise multiple comparison procedures (Dunn's Method); −Rapa $n = 47$, +Rapa $n = 52$). **F** Pseudo-colored representative images showing changes in mtCepia intensity without (blue) and with

(magenta) rapamycin treatment before and after agonist stimulation in R3-linker-expressing TKO HEK cells. Representative traces were obtained from the ROIs (yellow) shown in the upper images. **G** Representative mRFP and NAD(P)H images of a non-transfected and a linker-expressing cell (left). Traces from the same cells showing Cch and rotenone-induced changes in NAD(P)H fluorescence (middle) and different images showing the localization of NAD(P)H changes (right). **H** Agonist induced changes in the NAD(P)H autofluorescence with and without R3 linkage to the OMM (−Rapa $n = 49$, +Rapa $n = 46$). **I** Calculated the AUC of the Cch-induced NAD(P)H response. **J** Average traces showing the effect of 100 nM rapamycin treatment on the NAD(P)H level ($n = 27$). (*$p < 0.05$, **$p < 0.001$, Kruskal–Wallis One-way ANOVA on ranks, all pairwise multiple comparison procedures (Dunn's Method)). All box plots indicate median, 25th and 75th percentile (box) and 5th and 95th percentile (whiskers), and outliers (single points). Source data are provided for panels **B–E**, **G–J** as a Source Data file.

support local Ca²⁺ transfer to the mitochondria (Fig. 4A)? To investigate the effect of IP3R3 enrichment at the contacts on Ca²⁺ homeostasis, we simultaneously monitored changes in [Ca²⁺] in the cytosol and in the mitochondrial intermembrane space ($[Ca^{2+}]_{IMS}$) using Fura-2 and IMS-targeted RCaMP in TKO cells expressing FRB-IP3R3 and OMM-FKBP for 48 h. These experiments were performed in the absence of extracellular Ca²⁺, to eliminate non-ER-derived Ca²⁺ responses. The drug-induced linkage between IP3R3 and the OMM at the contact sites did not alter the cytosolic response to carbachol, but significantly increased the $[Ca^{2+}]_{IMS}$ (Fig. 4B, C). We performed the same measurement using matrix-targeted RCaMP together with Fura-2 in the cytosol (Fig. 4D). Again, we found no significant changes in the bulk cytosolic agonist response with or without rapamycin linkage, but a significant increase in $[Ca^{2+}]_m$ (Fig. 4E, F). Thus, recruitment of IP3R3 to the ERMCs enhanced IP3R-mediated ER-mitochondrial Ca²⁺ transfer after induction of the linkage (5 min), indicating that the OMM-linked IP3R3s were active upon capture at the contacts. Importantly, recruiting a generic ER membrane anchor unrelated to the IP3R (Sac1) to the OMM failed to improve the Ca²⁺ signal propagation to the mitochondria in WT HEK cells (Fig. S4B, C), highlighting the requirement for specifically trapping the IP3Rs.

We have demonstrated that capturing IP3Rs at ERMCs increased the Ca²⁺ mobilizing agonist-induced changes in $[Ca^{2+}]_{IMS}$ and $[Ca^{2+}]_m$ without affecting the global $[Ca^{2+}]_c$ response. To test potential downstream effectors of Ca²⁺ in the mitochondrial matrix, we recorded NAD(P)H autofluorescence, a readout of the activity of Ca²⁺-sensitive matrix dehydrogenases that are central to ATP production and the maintenance of mitochondrial redox balance. First, we analyzed the signal localization of NAD(P)H autofluorescence. We evaluated rapamycin-treated, IP3R3-linker transfected (mRFP expressing), and non-transfected cells on the same coverslip (Fig. 4G). Only the IP3R3-linker-expressing cells showed changes in NAD(P)H fluorescence after carbachol stimulation, but both responded to rotenone, a complex I inhibitor. By creating different images from before and after treatment with carbachol and rotenone, we established that the NAD(P)H signal increases overlapped with the mitochondrial area of the cells, and therefore we predominantly measured mitochondrial NAD(P)H changes. Next, we compared NAD(P)H responses in IP3R3-linker-expressing cells with and without rapamycin treatment. We found a significant increase in both the peak NAD(P)H fluorescence and the area under the curve (AUC) after Cch stimulation with rapamycin-induced linkage (Figs. 4H, I and S4A). NAD(P)H signals were normalized between the maximum (rotenone) and minimum after the addition of the uncoupling cocktail (FCCP and ATP synthase inhibitor oligomycin). The use of the immunophilin rapamycin at a low concentration (100 nM) to initiate dimerization between the artificial linkers did not on its own alter the NAD(P)H during 10 min of treatment (Fig. 4J).

To test the hypothesis that close contacts populated by IP3Rs are essential for efficient interorganellar Ca²⁺ signal propagation, we forced the mitochondria away from ER to the PM using a constitutive

OMM-PM linker. In this experiment, we used HeLa cells, which, despite having a similar prevalence of close ER-mitochondrial contacts as HEK cells, have efficient Ca²⁺ coupling after agonist (histamine) stimulation (Fig. S1A). We expressed a linker helix fused with OMM and PM targeting sequences (AKAP1 and CAAX, respectively) on opposite ends and monitored changes in $[Ca^{2+}]_c$ and $[Ca^{2+}]_m$ after the administration of histamine. An OMM-targeted construct was used as control (Fig. 5A). The OMM-PM linkage slightly increased the agonist-induced $[Ca^{2+}]_c$ response, but the magnitude of the $[Ca^{2+}]_m$ signal was reduced and the coupling time between the $[Ca^{2+}]_c$ and $[Ca^{2+}]_m$ was increased (Fig. 5B, C). The residual ER-mitochondrial Ca²⁺ communication in OMM-PM linker-expressing cells was likely a result of bringing the mitochondria close to subplasmalemmal ER segments where IP3Rs are already 'licensed' to respond to stimulation and/or the presence of some residual ERMCs. This experiment suggests that preventing the IP3Rs from juxtaposing with the OMM at ERMCs abrogates efficient local Ca²⁺ signal propagation to the mitochondria.

## Discussion

In this study of ERMCs structure and function, we employed an advance in the synthetic linker strategy to study the role of a particular, important physiological tethering species, the IP3R. With this technique, we were able to show that IP3Rs constantly move in and out of ERMCs and that trapping them at the ERMCs instantaneously improves the calcium signal propagation from the ER to the mitochondria leading to enhanced activation of energy metabolism. We expect a similar strategy could underpin the investigation of the dynamics of other natural tethers as well.

At the outset, we showed that local Ca²⁺ delivery from ER to the mitochondria was ineffective in HEK cells as compared to HeLa[30] or DT40 cells[25]. This was surprising since HEK cells express every IP3R isoform, show a robust $[Ca^{2+}]_c$ signal, and their ERMC ultrastructure is comparable to that of HeLa and DT40[25]. In DT40, an avian lymphoma cell line, IP3R2 was superior to IP3R1 and IP3R3 in Ca²⁺ transfer to the mitochondria[25]. To test whether this was also the case in human HEK cells, we used TKO cells in which all three IP3R isoforms were eliminated (ref. 31). In both acute and stable rescues, each IP3R isoform restored the $[Ca^{2+}]_c$ signal, but it was followed by only a slow and small $[Ca^{2+}]_m$ rise in the aggregate. However, in a fraction of the IP3R2-rescued TKO cells, the $[Ca^{2+}]_m$ rise closely followed the $[Ca^{2+}]_c$ signal, even with relatively low $[Ca^{2+}]_c$ elevations (Fig. 2B) and despite expression levels more comparable to endogenous levels in WT than the greatly overexpressed IP3R1 and IP3R2 (Fig. 2D). This result contrasts with previous findings that emphasized a discrete advantage for IP3R1 and IP3R3 in mammalian cells[23,22].

Regarding ER-mitochondrial structural coupling, TKO cells showed less tight contacts than WT cells, and rescue of the TKO with each isoform restored the WT phenotype (Fig. 2E). Taken together with the results on $[Ca^{2+}]_m$, restoration of the ERMCs by IP3Rs is insufficient to induce robust local Ca²⁺ coupling. We reason that a few tethers

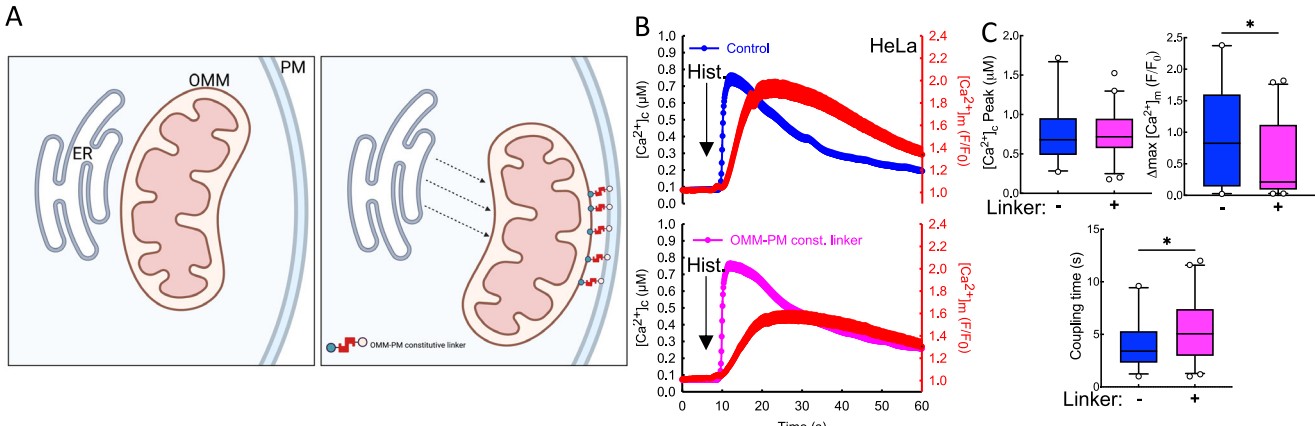

**Fig. 5 | ER-mitochondrial Ca²⁺ communication is disrupted by relocating mitochondria to the plasma membrane. A** Scheme depicting the disruption of ERMCs by the expression of PM-OMM constitutive linkers in WT HeLa cells. (Created in BioRender) **B** Average traces showing the effect of PM-OMM constitutive linker expression on ER-mitochondrial Ca²⁺ transfer after histamine stimulation compared to control cells expressing only OMM-mRFP (PM-OMM $n = 44$, OMM

only $n = 36$). **C** Comparison of the peak $[Ca^{2+}]_c$, $[Ca^{2+}]_m$ and coupling time of control and PM-OMM constitutive linker-expressing cells (*$p < 0.05$, Unpaired $t$-test; Two-tailed $p$ value; $[Ca^{2+}]_c$ peak $p = 0.8901$, $\Delta max[Ca^{2+}]_m$ $p = 0.0294$, Coupling time $p = 0.0314$). All box plots indicate median, 25th and 75th percentile (box), 5th and 95th percentile (whiskers), and outliers (single points). Source data are provided for panels **B**, **C** as a Source Data file.

might be enough to keep mitochondria close, but more channels are needed to produce the high $[Ca^{2+}]$ nanodomains that support local Ca²⁺ transfer. The IP3Rs securing the tight ERMCs likely belong to the immobile pool of IP3Rs that were described to mediate Ca²⁺ mobilization, at least during the Ca²⁺ puffs[32]. However, most IP3Rs are motile[26] and these might come into play to enhance the local Ca²⁺ flux and Ca²⁺ transfer to the mitochondria.

We found that the majority of each IP3R isoform is motile in HEK cells, and the overall mobility of the IP3R pool is unaltered by stimulation with a Ca²⁺ mobilizing agonist (Fig. 3A). To concentrate motile IP3Rs at ERMCs we adapted the drug-inducible ER-mitochondrial linker strategy[10]. This strategy and the robust lateral mobility of IP3Rs in the ER were validated by the enrichment of IP3R close to the mitochondria within minutes of linkage induction (Fig. 3C, D). Importantly, IP3R trapping occurred without creating new contacts, as shown by the lack of a change in the overall ER distribution. In this study, we employed FKBP-FRB heterodimerization by a rapamycin pulse, but this switch can be replaced by other drugs- or even light-inducible heterodimerization pairs[33]. In addition, we used constitutive synthetic linkers[8] to anchor mitochondria to the PM, away from ER, and showed an ensuing attenuation of the Ca²⁺ transfer from IP3R to the mitochondria in the inherently well-coupled HeLa cells (Fig. 5B). With further improvements in the technology, one could directly capture mobile IP3R at sites distant from the mitochondria. Similar strategies could be useful to study the structural and functional relevance of any ER or OMM tethering candidate that has lateral mobility in its host membrane. Furthermore, applications might expand to protein components of other organellar contacts.

$[Ca^{2+}]_{IMS}$ and $[Ca^{2+}]_m$ signals were greatly enhanced upon trapping IP3Rs close to the mitochondria, but there was no change in the $[Ca^{2+}]_c$ signal, confirming that the mtCU was exposed to higher $[Ca^{2+}]$ (Fig. 4B–E). This result suggests that motile IP3Rs recruited to ERMCs can be engaged in ER Ca²⁺ mobilization without any delay. This is important in the context of previous literature that emphasized the lack of Ca²⁺ release activity of motile receptors, and the need for 'licensing' for functional activation. Based on our findings, mobile IP3Rs either are quickly licensed upon immobilization[27,28] or bring "diffuse Ca²⁺ liberation"[34] to the ERMCs. The latter possibility implies that the Ca²⁺ release mode of the IP3Rs (punctate vs. diffuse)[34] could be an unrecognized factor controlling local Ca²⁺ transfer to the mitochondria. Since we used either the AKAP1 transmembrane domain or TOM70 as the OMM binding partner for the IP3R, our results also

support that VDAC does not have to be directly involved in the IP3R-dependent, ER-mitochondrial tether. Thus, further evaluation of the broadly accepted concept that an IP3R-GRP75-VDAC complex forms a Ca²⁺ tunnel between ER and mitochondria[22] might be needed. Because VDACs are the most abundant protein in the OMM, it is possible that anchoring IP3R at arbitrary OMM sites might be sufficient for Ca²⁺ transmission. The "Ca²⁺ tunnel" concept is also not entirely consistent with the earlier finding that a fast Ca²⁺ chelator could intercept the local IP3R-mitochondrial Ca²⁺ transfer[35]. Notably, our project was made possible by the distinctive impotence of ER-mitochondrial Ca²⁺ coupling in HEK cells, so further studies will be needed to determine what aspects of the IP3R distribution or regulation distinguish cell types that display robust IP3R-mitochondria Ca²⁺ signal propagation.

Lastly, we showed that activation of IP3Rs trapped at the OMM results in enhanced activation of the Ca²⁺-sensitive matrix dehydrogenases (Fig. 4H, I). This indicates that oxidative metabolism is responsive to fluctuations in the IP3R "Ca²⁺ messaging" towards the mitochondria. Thus, physiological rearrangements of the ERMCs and IP3Rs, including their documented alterations in many pathologies such as neurodegeneration, metabolic diseases, and malignancies, likely alter the coupling between cytoplasmic calcium signals and mitochondrial metabolism, a possibility that warrants further study.

## Methods

### Cells

Wild-type (WT), IP3R-deficient (TKO) HEK293 and rIP3R1, mIP3R2, rIP3R3, mIP3R2GS3 stable rescue TKO HEK293[18], and WT HeLa cells were grown in Dulbecco's modified Eagle's medium (DMEM), with 4.5 g/l glucose (Gibco, Cat# 11965118) supplemented with 10% FBS, 2 mM L-glutamine (Invitrogen), 100 units/ml penicillin/streptomycin (Lonza™, BioWhittaker™), in 5% $CO_2$ and 95% air at 37 °C.

### Constructs

mRFP-HA-FRB-IP3R3 were created by replacing the sequence of Cherry protein in a Cherry-rIP3R3 construct with the mRFP- and HA-tagged FRB-domain[36]. The construction resulted in a four amino acid linker (GSRA) between the FRB and the IP3R. Our efforts to create an HA-FRB-IP3R2 construct with the strategy used for IP3R3 failed.

### Transfection

Cells were transiently transfected using Lipofectamine 3000 (Thermo Fisher) with 1–3 μg total DNA following the manufacturer's protocol.

For Ca²⁺ measurements, TKO HEK cells were rescued with mammalian rIP3R1, mIP3R2, or mIP3R3. For mitochondrial [Ca²⁺] measurements, cells were transfected with matrix-targeted mtCepia3 (Addgene, Plasmid #58219) or mtGCaMP6f (kindly provided by Douglas Kim, Janelia Farm) or IMS-targeted IMS-RCamP (Cox8a-RCaMP1_035). For IP3R3 co-localization linker experiments, cells were transfected with mRFP-HA-FRB-IP3R3, AKAP1(34–63)-FKBP-YFP, and/or ER-ApoK1(lumen)-CFP. For IP3R3-linker Ca²⁺ experiments, cells were co-transfected with FRB-HA-IP3R3 and TOM70-FKBP or AKAP1(34–63)-FKBP constructs. For FRAP experiments, TKO HEK cells were rescued with mCherry-rIP3R1, mRFP-mIP3R2, or mRFP-mIP3R3 constructs. To facilitate inter-organellar linkage between the ER and the OMM, WT HEK cells were either co-transfected with mRFP-FRB-ER(Sac1)/mRFP-FRB-9x-FRB-ER(Sac1) and AKAP1(34–63)-FKBP-mRFP inducible linker pairs or AKAP1(34–63)-mRFP-(ER)Sac1/AKAP1(34–63)–9× Linker-mRFP-9x Linker-(UBC6) constitutive linkers.

### Linkage induction
Cells were treated with rapamycin for 10 min, followed by a washout and treatment with FK506 to prevent further co-localization. Cells were imaged shortly after this treatment.

### Western blot
Reagents for SDS-PAGE were obtained from Bio-Rad. Cells were harvested and washed twice with ice-cold PBS then solubilized in cell lysis buffer containing 10 mM Tris-HCl, 10 mM NaCl, 1 mM EGTA, 1 mM EDTA, 1 mM NaF, 20 mM $Na_4P_2O_7$, 2 mM $Na_3VO_4$, 1% Triton X-100 (v/v), 0.5% sodium deoxycholate (w/v), and 10% glycerol with a mixture of protease inhibitors (Roche). Lysates were incubated on ice for 30 min, then cleared by centrifugation at 16,000×$g$ for 10 min at 4 °C, then transferred into fresh tubes. Protein concentrations were quantified with a DC protein assay kit (Bio-Rad). Protein samples were resolved on 5–7.5% SDS-PAGE gels and then transferred to nitrocellulose membranes (0.45 µm, Bio-Rad). Rabbit polyclonal antibodies recognizing IP3R1 (CT1) (1:5000) raised against the C-terminal 19 amino acid (aa) residues of rat IP3R1 and IP3R2 (NT2) (1:200) raised against N-terminal 320–338 aa residues in mouse IP3R2 (custom made, Pocono Rabbit Farms and Laboratories), mouse monoclonal antibody against 22–230 aa residues of human IP3R3 (1:1000, BD Transduction laboratories Cat# 610312), Calnexin rabbit polyclonal antibody (1:1000, Enzo Cat# ADI-SPA-860), and mouse monoclonal Flag antibody (1:5000, Sigma Cat# F1804- 200UG) were used for immunoblotting. Corresponding mouse and rabbit HRP-linked secondary antibodies were obtained from Cell Signaling (1:5000, Anti-mouse IgG, HRP-linked Antibody Cat#7076, Anti-rabbit IgG, HRP-linked Antibody Cat#7074). Proteins were detected with chemiluminescence using an HRP substrate for CCD imaging (Azure Biosystems). Elements of the mitochondrial uniporter complex were detected using anti MCU (1:1000, Bethyl, A300-BL19220), anti MICU1 (1:1000, Sigma, HPA037480), anti MICU2 (1:1000, Bethyl, A300-BL19212), anti EMRE (1:1000, Bethyl, A300-BL19208), anti HSP70 (1:1000, Invitrogen, PA5-17612) and visualized by LiCor secondary antibodies.

### Fluorometry
Saponin-permeabilized HEK cells were resuspended in 1.5 ml of intracellular medium containing 120 mM KCl, 10 mM NaCl, 1 mM $KH_2PO_4$, 20 mM Tris-HEPES at pH 7.2, and supplemented with proteases inhibitors (leupeptin, antipain, pepstatin, 1 µg/ml of each), 2 mM MgATP, 2 µM thapsigargin (Enzo) and maintained in a stirred thermostated cuvette at 35 °C. Fura-2-FF (1 µM, Teflabs) was added to the suspension to monitor [Ca²⁺] in the medium, and 1.5 µM TMRM (Invitrogen) to monitor corresponding changes in the $\Delta\psi_m$. Assays were performed in the presence of 20 µM CGP-37157 (Enzo) and 2 mM succinate. Fluorescence was recorded in a multi-wavelength excitation dual-wavelength emission fluorimeter (DeltaRAM, PTI) using 340–380 nm

excitation and 500-nm emission for fura-2-FF and 545 nm excitation and 580 nm emission for TMRM. Five data triplets were obtained per second. Ca²⁺-clearance was calculated after the addition of 20 µM Ca²⁺ pulse and complete depolarization (maximum de-quench of TMRM fluorescence) was elicited using the protonophore FCCP (2 µM). The Fura signal was calibrated at the end of each measurement, adding 1 mM CaCl₂, followed by 10 mM EGTA/Tris, pH 8.5.

### Transmission electron microscopy
HEK cells were fixed using 2% glutaraldehyde, then with 1% $OsO_4$ and stained with 0.5% uranyl acetate, pelleted in 2% agarose (Sigma–Aldrich, Type IX ultralow gelling temperature), dehydrated in a dilution series of acetone/ water, and embedded in Spurr's resin (Electron Microscopy Sciences). The sections were examined with a JEOL JEM1010 TEM fitted with a side-mounted AMT XR- 50 5Mpx CCD camera or an FEI Tecnai 12 TEM fitted with a bottom-mounted AMT XR-111 10.5 Mpx CCD camera. Whole-cell images were taken with ×6700–14,000 (JEOL) or ×4400–6500 (FEI) direct magnification. Mitochondria were imaged using ×40,000 (JEOL, 1.5 nm/px resolution) or ×15,000 (FEI, 1.3 nm/ px resolution) magnification. Interfaces between ER and mitochondria were analyzed using a custom ImageJ script: https://sites.imagej.net/MitoCare/.

For immuno-EM experiments, TKO HEK cells were sorted by flow cytometry (Aria) for double positives of mRFP-HA-FRB-IP3R3 and OMM-FKBP-YFP. Sham-transfected control TKO cells were treated the same way as the transfected cells. An equal count of sorted cells was plated on 14 mm plastic coverslips (Thermanox) for 6 h before treatment and processing. Cells were fixed with 3% paraformaldehyde followed by quenching, permeabilization, blocking, and labeling with primary antibody (dilution: 1:500, Abcam, Cat#ab9110), followed by secondary blocking and incubating with secondary antibody anti-rabbit IgG Nanogold (dilution: 1:50 Nanoprobes, Cat#2001–0.5 ml). Samples were dehydrated in a graded series of alcohol and embedded in Durcupan (Electron Microscopy Sciences). After polymerization, ultrathin sections (65–80 nm) were prepared using ultramicrotome (Leica UTC, Diatome knife, Diatome) and studied with FEI Tecnai 12 electron microscope. Images were taken at ×3200–15,000 magnification.

### Epifluorescence imaging
**Ca²⁺ measurements.** HEK and HeLa cells plated on Poly-D-Lysine coated round (25 mm) coverslips (Fisher Cat#) were loaded with 2 µM fura-2/AM (Invitrogen) in a serum-free extracellular medium (ECM, 121 mM NaCl, 5 mM NaHCO₃, 10 mM Na-HEPES, 4.7 mM KCl, 1.2 mM KH₂PO₄, 1.2 mM MgSO₄, 2 mM CaCl₂, and 10 mM glucose, pH 7.4) containing 2% BSA (Roche) in the presence of 0.003% Pluronic F-127 and 150 µM sulfinpyrazone for 15 min at 37 °C to measure changes in the [Ca²⁺]_c. After dye-loading, cells were washed with fresh ECM containing 0.25% BSA and transferred to the thermostated stage (37 °C) of the microscope. For the epifluorescence Ca²⁺ imaging, the ImagEM EM-CCD camera (Hamamatsu) was fitted to an Olympus IX81 microscope with Lambda DG4 light source operated by Metamorph (Molecular Devices). The imaging setups used the UV-optimized Olympus UAPO/ 340 ×40/1.35 N.A. oil immersion objective. For simultaneous measurements of [Ca²⁺]_c and [Ca²⁺]_m or [Ca²⁺]_IMS, Fura-2 fluorescence was recorded at 340 and 380 nm, the mitochondrial matrix-targeted Cepia3 or 485/15 nm or IMS-targeted RCaMP at 577/20 nm excitations, using dual-band dichroic and emission filters (Chroma, set 73100 or customized set 59022, respectively). Images were acquired every 0.2 s or 0.33 s. Fura-2 was calibrated in vitro by adding 1 mM CaCl₂, followed by 10 mM EGTA/Tris, pH 8.5. [Ca²⁺]_c was determined by the following formula: $[Ca^{2+}] = K_d \times (Sf_2/Sb_2) \times (R - Rmin)/(Rmax - R)$, where $K_d$ is the Ca²⁺-dissociation constant (0.224 µM), R is the ratio of the fluorescence intensities at 340/380 nm excitation, Rmin and Rmax are the fluorescence ratios in Ca²⁺-free and Ca²⁺-saturated conditions, respectively,

$Sf_2/Sb_2$ is the ratio of fluorescence intensities measured at 380 nm excitation in $Ca^{2+}$-free (f)/$Ca^{2+}$-saturated ($Ca^{2+}$-bound, b) conditions.

**NAD(P)H autofluorescence measurements.** Leica DMI 6000B epifluorescence microscope, Olympus UAPO/340 40x/1.35 N.A. oil immersion objective and a ProEM 1024 EM-CCD camera (Princeton Instruments) was used to measure NAD(P)H autofluorescence. A high-pressure Xenon lamp was used for illumination with 350/20 nm excitation, 400 nm long-pass dichroic and 460/50 nm emission filters. Hardware synchronization and image acquisition was controlled through a Windows PC using custom software (Spectralyzer).

## Confocal imaging

Zeiss LSM 880 confocal system equipped with Airyscan detector, using 488, 561, and 633-nm laser lines for green, red, and far-red fluorescent probes, respectively. Images were taken through a ×63 oil objective (numerical aperture [NA] 1.40). Images were captured using Carl Zeiss ZEN 2.3 software package (Carl Zeiss Microscopy GmbH 1997–2015) and were processed using ImageJ (Wayne Rasband, National Institutes of Health, USA).

FRAP experiments were performed on a Zeiss LSM780 NLO system using a 63x oil immersion objective (PlanApo, 1.4 N.A.). DPSS laser at 561 nm was used for the excitation of mRFP/mCherry. Circular regions 5 μm in diameter were photobleached by pulsed Ti:Sapphire laser (Coherent Chameleon Vision II) at 720 nm to achieve rapid, extensive bleaching. Images were acquired at 1 frame/s using Carl Zeiss ZEN 2.3 software package (Carl Zeiss Microscopy GmbH 1997–2015) and were processed using ImageJ (Wayne Rasband, National Institutes of Health, USA).

## STORM Imaging

3D super-resolution microscopy was performed using a Vutara 352 nanoscope (Bruker). Briefly, two focal planes were projected onto the sensor of an ORCA Flash v.2 sCMOS camera (Hamamatsu) through an Olympus UPlanSApo ×60/1.2 N.A. water-immersion objective. The point spread functions for the wavelengths used were calibrated and aligned using 100 nm Tetraspeck microspheres (Thermo Fisher Scientific). Laser illumination at 640 and 561 nm were used for stochastic imaging of AlexaFluor 647 and CF568 (Biotium), respectively, and 405 nm light was used to recover portions of CF568 from their dark state. The two colors were recorded sequentially (AF647, then CF568) and several thousand frames were acquired for each fluorophore. Cells were plated, fixed and IP3R-HA were labeled with HA (dilution: 1:400, Abcam ab9110) primary then AF647 (dilution: 1:200, Molecular Probes A21239) and mtHSP70 (JG1) (dilution: 1:100, Invitrogen MA3-028) primary then with CF568 Goat Anti-Rabbit IgG (dilution: 1:200, Biotium 20103-1) secondary antibodies. Imaging was performed in freshly prepared buffer for oxygen- depleting and reducing conditions: 20 mM cysteamine, 1% 2-mercaptoethanol (v/v), 169 AU/ml glucose oxidase and 1400 AU/ml catalase in 50 mM Tris-HCl, 10 mM NaCl, and 10% (w/v) glucose (pH 8.0). Calibration, alignment, acquisition, and localization were all done in the Vutara SRX software.

## Statistics

Statistical testing was performed using either SigmaPlot (Systat Software, Inc.), Prism (GraphPad Software, LLC.), or Excel (Microsoft). For comparisons between groups, first, it was determined whether the data were normally distributed using the Shapiro–Wilk test (SigmaPlot). If data were normally distributed, one-way ANOVA was used with post hoc Holm–Sidak test for pairwise comparisons or an unpaired *t*-test with two-tailed *p* values. If not, Mann–Whitney rank sum test (two groups) or Kruskal–Wallis one-way ANOVA on ranks (three or more groups) with post hoc Dunn's test was used. $p < 0.05$ was considered significant for all tests. All graphs were created in SigmaPlot 12 or Prism 9 and were combined with the images in Microsoft PowerPoint and Adobe Illustrator. Illustrations were created with Adobe Illustrator and BioRender (BioRender.com).

Every element of each graph and image was created by the Authors. No previously created elements were used.

## Reporting summary

Further information on research design is available in the Nature Research Reporting Summary linked to this article.

## Data availability

The datasets generated during and/or analyzed during the current study are available from the corresponding author on reasonable request. Source data are provided with this paper.

## Code availability

Custom image analysis software (Spectralyzer) and macros prepared for ImageJ are available from the corresponding author upon reasonable request. The ImageJ plugin for analysis of ER-mitochondrial interfaces in TEM images is available from the updated site: http://sites.imagej.net/MitoCare/.

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

## Acknowledgements

M.K. was supported by the Rosztoczy Foundation Scholarship. A.B. was supported by Hungarian State Eotvos Fellowship MÁEÖ2016_24 from the Tempus Public Foundation (Hungary) and a János Bolyai Research Scholarship of the Hungarian Academy of Sciences (BO/00103/20/8). This study was funded by NIH grants DK051526 and GM059419 (G.H.), ES025672 (G.H. and G.C.), DK103558 (G.H.), and DE014756 (D.I.Y.).

## Author contributions

Conceptualization: G.H., G.C., D.W., D.I.Y., and M.K.; Investigation: M.K., A.B., Z.N., D.W., A.G., and E.B.; Generating reagents: D.I.Y. and P.V., Writing: G.H., M.K., and D.W.; Funding acquisition: G.H., G.C., D.I.Y., A.B., and M.K.

## Competing interests

The authors declare no competing interests.
