## [Peer Review File · Nature Communications]

Capture at the ER-Mitochondrial Contacts Licenses IP3 receptors to Stimulate Local Ca²⁺ Transfer and Oxidative MetabolismReviewers' Comments:

Reviewer #1:

Remarks to the Author:

The primary findings of this study are novel and important. The authors have characterised the prominent role of IP3Rs in the formation of ER-Mitochondria junctions/contacts (ERMCS). Notably, (and unexpectedly) this ability was shared by all 3 types of IP3Rs. Furthermore, 'pore-dead' IP3R2 mutant was able to induce contacts between the organelles. This finding was further extended by characterising the consequence of capturing IP3R in ERMCS. The authors demonstrated that the interaction between IP3Rs and the outer mitochondrial membrane strongly potentiated agonist-induced Ca²⁺ transfer into mitochondria and downstream upregulation of NAD(P)H production.

I have only few minor comments:

It could be advantageous to demonstrate FAD fluorescence changes associated with capturing IP3Rs at the outer mitochondrial membrane. This could reconfirm the mitochondrial origin of the NAD(P)H changes.

There is a problem with labelling on the Figure 2 A (see the upper line of labelling). Please correct.

Lines 294 and 295 "magnitude of the [Ca²⁺]_m signal and the coupling time between the [Ca²⁺]_c and [Ca²⁺]_m was drastically reduced (Fig4J,K)". [Ca²⁺]_m has indeed reduced, whilst the coupling time has increased (as expected) but not "drastically". Please correct.

Tethering mitochondria to the plasma membrane could bring these organelles into close proximity with subplasmalemmal ER strands and IP3Rs 'licensed' by a different mechanism. Please consider discussing this point.

Reviewer #2:

Remarks to the Author:

The manuscript submitted by Katona et al investigates how the mobility of IP3Rs impacts on the generation of ER-mitochondria coupling. The authors took advantage of a cellular model (HEK) where ER-mito tethering is somewhat intrinsically inefficient, due to unknown reasons. They were able to study and visualize the recruitment of all IP3Rs to ER-mito contacts along with their functionality in terms of Ca²⁺ coupling between the two organelles. Overall, the paper is technically sound and logically presented. The findings are interesting and provide novel insights to the molecular understanding of contacts sites between ER and mito, possibly applicable to other membrane contacts as well. I have just few comments to improve the manuscript:

- Please provide expression data of MCU complex components also after re-introduction of IP3Rs in TKO cells (possibly at both 48h and 120h after re-expression of IP3Rs)
- The manuscript shows a nice systematic overview of all IP3Rs isoforms. However, drug induced linkage between IP3R and OMM has been performed only for R3 (Fig 4BC). Based on previous data, one should expect no differences, but it would be nice to have direct evidence for that
- Fig 2A: please fix the legends in the upper panels. It should be "TKO + R1", "TKO + R2" and "TKO + R3", where "TKO + R1" is invariably indicated.

Reviewer #1 (Remarks to the Author):

The primary findings of this study are novel and important. The authors have characterised the prominent role of IP3Rs in the formation of ER-Mitochondria junctions/contacts (ERMCS). Notably, (and unexpectedly) this ability was shared by all 3 types of IP3Rs. Furthermore, 'pore-dead' IP3R2 mutant was able to induce contacts between the organelles. This finding was further extended by characterising the consequence of capturing IP3R in ERMCS. The authors demonstrated that the interaction between IP3Rs and the outer mitochondrial membrane strongly potentiated agonist-induced Ca²⁺ transfer into mitochondria and downstream upregulation of NAD(P)H production.

I have only few minor comments:

We appreciate the Reviewer's positive comments and address her/his useful suggestions below.

It could be advantageous to demonstrate FAD fluorescence changes associated with capturing IP3Rs at the outer mitochondrial membrane. This could reconfirm the mitochondrial origin of the NAD(P)H changes.

To test the origin of the NAD(P)H changes we have analyzed the subcellular spatial pattern of the IP3-linked fluorescence increase. We found that the increase appeared in perinuclear distribution consistent with the mitochondrial distribution. Furthermore, the increase effectively colocalized with the distribution of the mitochondria-targeted RFP. We show the images and graphs in Fig4I in the revised ms. (Ln280-290)

We also attempted to record FAD autofluorescence, but in the HEK cells the basal fluorescence was very low, such that rotenone did not produce a reliable decrease from baseline. We tried to use substrates (acetoacetate and lactate/pyruvate) to elevate the baseline, but they were ineffective in these cells, and we were not able to visualize any decrease following agonist stimulation. We were only able to measure an increase with uncoupler which was reversed by subsequent rotenone treatment (not shown).

There is a problem with labelling on the Figure 2 A (see the upper line of labelling). Please correct.

The typos have been fixed.

Lines 294 and 295 "magnitude of the [Ca²⁺]_m signal and the coupling time between the [Ca²⁺]_c and [Ca²⁺]_m was drastically reduced (Fig4J,K)". [Ca²⁺]_m has indeed reduced, whilst the coupling time has increased (as expected) but not "drastically". Please correct.

The text has been corrected as follows:

"The OMM-PM linkage slightly increased the agonist induced [Ca²⁺]_c response, but the magnitude of the [Ca²⁺]_m signal was reduced and the coupling time between the [Ca²⁺]_c and [Ca²⁺]_m was increased." (ln 303-308).

Tethering mitochondria to the plasma membrane could bring these organelles into close proximity with subplasmalemmal ER strands and IP3Rs 'licensed' by a different mechanism. Please consider discussing this point.

Discussion was added as follows:

"The OMM-PM linkage slightly increased the agonist induced [Ca²⁺]_c response, but the magnitude of the [Ca²⁺]_m signal was reduced and the coupling time between the [Ca²⁺]_c and [Ca²⁺]_m was increased (Fig5B,C). The residual ER-mitochondrial Ca²⁺ communication in OMM-PM linker expressing cells was likely a result of bringing the mitochondria close to subplasmalemmal ER segments where IP3Rs are already 'licensed' to respond to stimulation and/or the presence of some residual ERMCS." (lns303-308).

Reviewer #2 (Remarks to the Author):

The manuscript submitted by Katona et al investigates how the mobility of IP3Rs impacts on the generation of ER-mitochondria coupling. The authors took advantage of a cellular model (HEK) where ER-mito tethering is somewhat intrinsically inefficient, due to unknown reasons. They were able to study and visualize the recruitment of all IP3Rs to ER-mito contacts along with their functionality in terms of Ca²⁺ coupling between the two organelles. Overall, the paper is technically sound and logically presented. The findings are interesting and provide novel insights to the molecular understanding of contacts sites between ER and mito, possibly applicable to other membrane contacts as well. I have just few comments to improve the manuscript:

We are grateful for the Reviewer's the constructive criticism and positive comments.

- *Please provide expression data of MCU complex components also after re-introduction of IP3Rs in TKO cells (possibly at both 48h and 120h after re-expression of IP3Rs)*

We have performed the proposed biochemistry experiments and included the results in FigS2. No change in any MCU complex component was detected during the re-expression of the IP3Rs. (ln180-182).

- *The manuscript shows a nice systematic overview of all IP3Rs isoforms. However, drug induced linkage between IP3R and OMM has been performed only for R3 (Fig 4BC). Based on previous data, one should expect no differences, but it would be nice to have direct evidence for that*

We also set out to create linker constructs for the other IP3R isoforms, particularly for IP3R2. The authors of this ms include outstanding molecular biologists in the Yule and Varnai laboratories, who have productive track record with IP3R constructs, but they were unable to create the needed constructs. We have mentioned this problem in the Methods. (ln403-404).

- *Fig 2A: please fix the legends in the upper panels. It should be "TKO + R1", "TKO + R2" and "TKO + R3", where "TKO + R1" is invariably indicated.*

The typos have been fixed.

Reviewers' Comments:

Reviewer #1:

Remarks to the Author:

All my suggestions have been addressed satisfactorily.

Reviewer #2:

Remarks to the Author:

The authors successfully addressed my (few) concerns.